# Urogenital schistosomiasis among pre-school and school aged children in four districts of north western Tanzania after 15 years of mass drug administration: Geographical prevalence, risk factors and performance of haematuria reagent strips

**Humphrey D. Mazigo** [1,2]*, **Upendo J. Mwingira** [3,4], **Maria M. Zinga** [1], **Cecilia Uisso** [3], **Paul E. Kazyoba** [5], **Safari M. Kinung'hi** [6], **Francesca Mutapi** [2,7]

**1** Department of Medical Parasitology, School of Medicine, Catholic University of Health, and Allied Sciences, Mwanza, Tanzania, **2** NIHR Global Health Research Unit Tackling Infections to Benefit Africa, University of Edinburgh, Ashworth Laboratories, King's Buildings, Edinburgh, United Kingdom, **3** National Neglected Tropical Diseases Control Programme, National Institute for Medical Research, Barack Obama Drive, Dar-Es-Salaam, Tanzania, **4** RTI International, Washington, United State of America, **5** National Institute for Medical Research, 3 Barack Obama Drive, Dar-Es-Salaam, Tanzania, **6** National Institute for Medical Research, Mwanza Centre, Mwanza, Tanzania, **7** Institute of Immunology and Infection Research, University of Edinburgh, Ashworth Laboratories, King's Buildings, Edinburgh, United Kingdom

* humphreymazigo@gmail.com

## Abstract

### Background

Urogenital schistosomiasis remains as a public health problem in Tanzania and for the past 15 years, mass drug administration (MDA) targeting primary school children has remained as the mainstay for its control. However, after multiple rounds of MDA in highly risk groups, there are no data on the current status of *Schistosoma haematobium* in known endemic areas. Furthermore, the performance of commonly used diagnostic test, the urine reagent strips is not known after the decline in prevalence and intensities of infection following repeated rounds of treatment. Thus, after 15 of national MDA, there is a need to review the strategy and infection diagnostic tools available to inform the next stage of schistosomiasis control in the country.

### Methods/Findings

A analytical cross-sectional study was conducted between October and November, 2019 among pre-school (3-5years old) and school aged children (6–17 years old) living in four (4) districts with low (<10%) and moderate (10%-<50%) endemicity for schistosomiasis as per WHO classification at the start of the national control programme in 2005/06, with mean prevalence of 20.7%. A total of 20,389 children from 88 randomly selected primary schools participated in the study. A questionnaire was used to record demographic information. A

**Data Availability Statement:** All relevant data are within the manuscript and its Supporting Information files.

**Funding:** This research was commissioned by the National Institute for Health Research (NIHR) Global Health Research programme (16/136/33) using the UK Aid from the UK Government. The grant supported postdoctoral training of HDM and UJM. The funders had no role in study design, data collection and analysis, decision to publish, or preparation of the manuscript.

**Competing interests:** The authors have declared that no competing interests exist.

single urine sample was obtained from each participant and visually examined for macro-haematuria, tested with a dipstick for micro-haematuria, to determine blood in urine; a marker of schistosome related morbidity and a proxy of infection. Infection intensity was determined by parasitological examination of the urine sample for *S. haematobium* eggs. Overall, mean infection prevalence was 7.4% (95%CI: 7.0–7.7, 1514/20,389) and geometric mean infection intensity was 15.8eggs/10mls. Both infection prevalence (5.9% *versus* 9%, $P<0.001$) and intensity (t = -6.9256, $P<0.001$) were significantly higher in males compared to females respectively. Light and heavy infections were detected in 82.3% and 17.7% of the positive children respectively. The prevalence of macrohaematuria was 0.3% and that of microhaematuria was 9.3% (95%CI:8.9–9.7). The sensitivity and specificity of the urine reagent strip were 78% (95%CI: 76.1–79.9) and 99.8% (95%CI: 99.7–99.9). Having light ($P<0.001$) and heavy infection intensities ($P<0.001$) and living in the study districts increased the odd of having microhaematuria. Predictors of *S. haematobium* infection were being male ($P<0.003$), microhaematuria ($P<0.001$), and living in the three study districts ($P<0.001$) compared to living at Nzega district.

## Conclusion

The findings provide an updated geographical prevalence which gives an insight on the planning and implementation of MDA. Comparing with the earlier mapping survey at the start of the national wide mass drug administration, the prevalence of *S. haematobium* infection have significantly declined. This partly could be attributed to repeated rounds of mass drug administration. The urine reagent strips remain as a useful adjunct diagnostic test for rapid monitoring of urogenital schistosomiasis in areas with low and high prevalence. Based on prevalence levels and with some schools having no detectable infections, review of the current blanket mass drug administration is recommended.

## Author summary

Mass drug administration using praziquantel against *Schistosoma haematobium*, the causative agent of urogenital schistosomiasis, is the main intervention measure in Tanzania and single annual rounds of MDA has been implemented consecutively for 15 years in north-western part of the country. Therefore, it is highly recommended to assess the current status of S. *haematobium* infection in these areas. Re-assessment will help to identify remaining pockets of the disease and geographical location of at-risk population. This will allow data-driven improvements of MDA strategies in areas remaining with high prevalence, distribution of the limited available drugs and funds to support preventive mass chemotherapy. This study determined the prevalence and *S. haematobium* intensity of infections after 15 years of mass preventive chemotherapy. Furthermore, the study assessed the performance of urine reagent strip at this time when the infection prevalence and intensities have declined following mass treatment. Overall, comparing with the baseline prevalence prior to initiation of countrywide mass preventive chemotherapy in 2005/06, though not uniformly, the prevalence of *S. haematobium* infection have significantly declined in all the four-study district. However, some pockets of infections remain in some of the districts. In addition, even after 15 years of intensive preventive mass

chemotherapy, the performance of urine reagent strips is acceptable and can continue to be used for rapid assessment and monitoring the impact of mass preventive chemotherapy on *S. haematobium* infection.

## Background

After Nigeria, Tanzania's national prevalence of 52% of schistosomiasis [1,2] is the highest on the African continent, with two -thirds of the schistosomiasis cases being due to *Schistosoma haematobium* [1,2]. *S. haematobium* is endemic throughout the country but its transmission is focalized and heterogeneous in nature [3]. Mass drug administration (MDA) using praziquantel is the main strategy for controlling urogenital schistosomiasis in Tanzania and the main focus is on primary school children [4]. Over 15 years, multiple rounds of MDA using praziquantel have been conducted in the country and are reported to have resulted into declining in prevalence of infection [5–8]. However, not all treated areas have achieved a uniform decline in prevalence and infection intensity [7,9]. In some districts, and some village(s), levels of *S. haematobium* infection (prevalence and mean intensity) have remained consistent despite several years of treatment rounds [7,9]. Thus, before planning and implementing the next MDA rounds, it is important to re-assess the *S. haematobium* infection levels in districts which were categorized has having low (prevalence of *S. haematobium* <10%) and moderate (prevalence of *S. haematobium* ≥10%—<50%) endemicity by previous mapping [10]. Re-assessment will identify remaining pockets of the disease and geographical location of at-risk population. This will allow data-driven improvements of MDA strategies in areas remaining with high prevalence, distribution of the limited available drugs and funds to support preventive mass chemotherapy.

In schistosomiasis endemic areas, after repeated rounds of preventive chemotherapy, the majority of the treated individuals present with either no detectable *S. haematobium* eggs or with light infection intensities [11,12], this imposes challenge for accurate diagnosis [12]. Sensitivity and specificity of the diagnostic tools may be affected by the changes in infection levels following repeated rounds of preventive chemotherapy [11]. Thus, after 15 years of MDA, the field performance of the commonly used diagnostic techniques for *S. haematobium* need to be re-assessed. The standard diagnostic test for *S. haematobium* infection is the microscopic quantification of eggs processed using urine filtration technique [13]. The performance of this technique is described elsewhere [11,14]. In urogenital schistosomiasis endemic communities' presence of blood in urine or reported visible blood in urine (macrohematuria) are used as indicators of infection [15–17]. In these communities, a standardized questionnaire asking for visible haematuria has been used to identify cases of urogenital schistosomiasis [17,18]. A reagent urine strip(s) is another recommended proxy for diagnosis of *S. haematobium* infection and it detects micro-haematuria [16,17]. However, its performance after multiple rounds of preventive chemotherapy is not known and we hypothesize that multiple rounds of treatment affect its performance when compared with urine filtration technique [11]. Thus, its performance after multiple rounds of preventive chemotherapy needs reassessment.

In that context, the current study aimed at (i) assessing the prevalence and intensity of infection of *S. haematobium* among pre-and-school aged children (ii) assessing the performance (sensitivity and specificity) of urine reagent strips as an indirect test for *S. haematobium* using an egg microscopy test as standard diagnostic test and (iii) secondarily, assessed the risk factors for *S. haematobium* infections after 15 years of repeated rounds of preventive chemotherapy.

## Method

### Ethics statement

Ethical approval for this study was sought from the joint Institution review board of the National Ethical Committee, under the National Institute for Medical Research (certificate number NIMR/HQ/R.8a/Vol.1X/3074). The study also sought permission from the Regional and District Administrative Authorities of region and study districts.

Written informed consent was obtained from the parents/guardians of the children. An assent form was also used for children aged 9–17 years. Participants were only included in the study after submission of written informed and assent forms. For confidentiality purposes, all clinical and demographical data from the study participants were kept in a closed cabinet and all participants were identified using codes. All children identified being infected with *S. haematobium* were treated using PZQ (40mg/kg) according to WHO recommendation [19].

### Study areas

This study was conducted in north-western Tanzania, specifically in Nzega, Itilima, Bariadi and Shinyanga rural districts (Fig 1), these are areas with low to moderate endemicity for *S.*

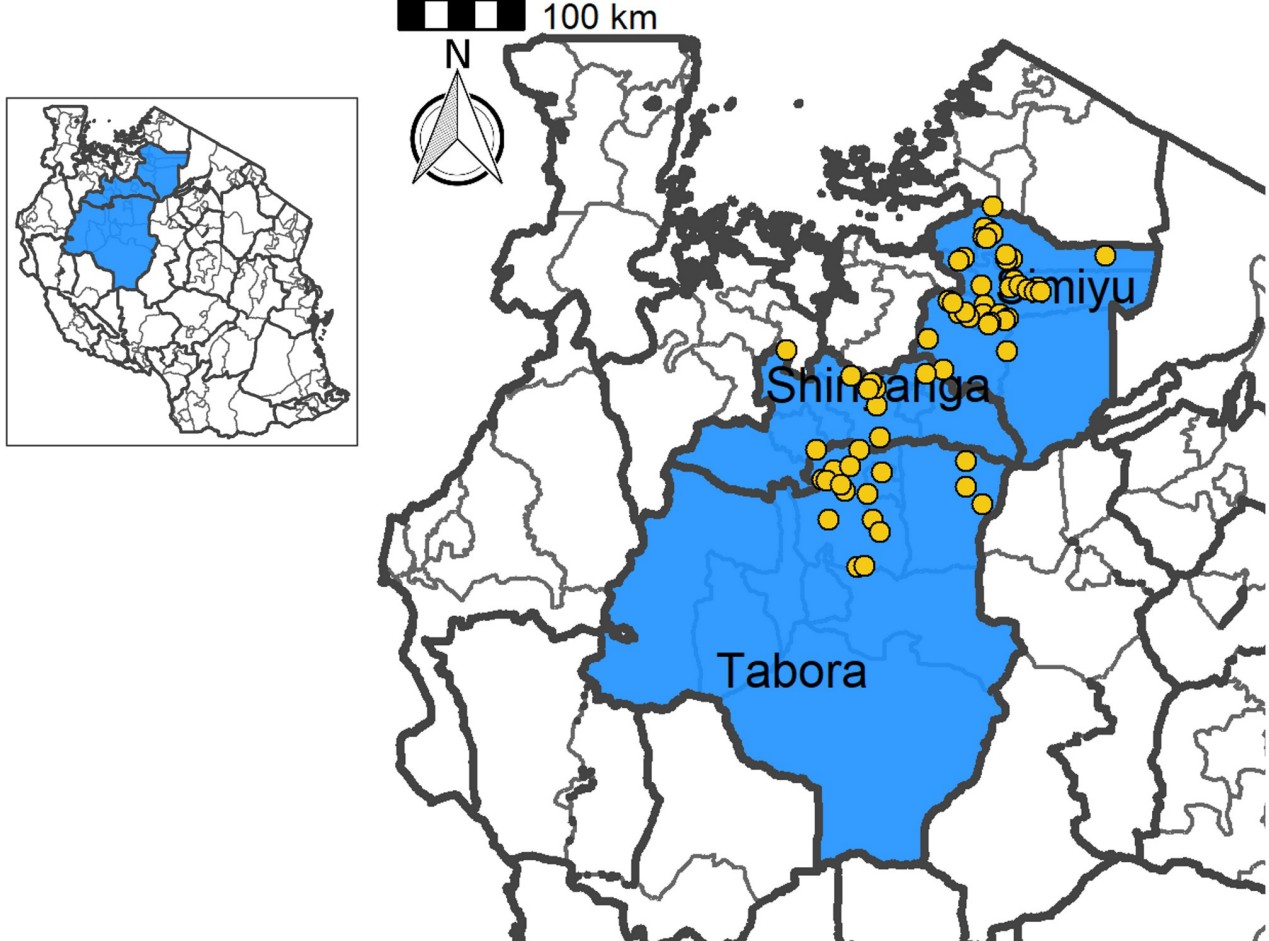

**Fig 1. Study schools at Itilima, Nzega, Shinyanga rural and Bariadi districts, north-western Tanzania (credit to https://gadm.org/maps/TZA_1. html).**

*haematobium* infection [5]. These districts have different levels of *S. haematobium* endemicity (prevalence) and continue to experience high transmission in some villages despite repeated rounds of MDA [2,20]. The pre-MDA schistosomiasis mapping between 2004/2005, categorized the selected districts into low to moderate endemicity with prevalence ranging from 10% - 49.9% [10]. During this survey, a total of 5,201 school aged children from 86 schools selected from the current study districts were involved in the study. The overall prevalence of *S. haematobium* was 20.7% (1075/5201) [10]. The prevalence of *Schistosoma mansoni*, *Ascaris lumbricoides*, *Trichuris trichiura* and hookworm were 0.8%, 0.28%, 0.13% and 31.8% [10]. Data from the National Strategic plan for controlling schistosomiasis of 2010 using the Red Urine Questionnaire indicated that the overall prevalence of *S. haematobium* infection of >50% in the same districts [5,6]. The selected districts have conducive environmental conditions for transmission of *S. haematobium* [2]. The four districts experience annual rainfall ranges between 930–1,200mm with the long rainy seasons experienced between January and May. The temperature ranges between 25°C–28°C with an annual average temperature of 26.5°C. The topography of these districts is mainly characterized with black soil suitable for rice cultivation and the terrain is characterized with water bodies. In both districts, the inland areas are covered with seasonal rivers and streams which drain in inland man-made and natural water bodies. These inland water bodies maintain transmission of *S. haematobium* at the end of the long rain season or during the dry season [21]. The transmission of *Schistosoma mansoni* and soil-transmitted helminths is very low and the infections levels ranges from 0.1–5% [2,22]. The economic activities of the inhabitants are mainly farming, livestock keeping and small-scale business. Mass drug administration using praziquantel drug (PZQ) is the main strategy for controlling of schistosomiasis, with the main targets being the school aged children (SAC). From 2000, SAC living in these districts have been receiving a single annual round of MDA using PZQ and the last MDA was conducted in 2017 and data on SAC compliance to treatment remain unknown. In 2018, SAC did not receive treatment.

## Study design, inclusion, and exclusion criteria

This study was conducted between September-October, 2019 and used an analytical cross-sectional study design to collect data from 88 primary schools located in North-western Tanzania. The selection of the district was based on the previous reports which indicate that the districts were endemic for *S. haematobium* [5,6,10,20]. The study included pre-school (3–5 years) and school aged children aged 6–17 years of age attending the randomly selected schools in the four districts. The inclusion criteria were age between 3–17 years, living in the study villages and attending the selected schools, having no history of using PZQ in the past 6 months prior to the study based on MDA reports available at the school, presence of parents/guardians for pre-school children and presenting a signed informed consent from guardians/parents and assent form for children aged 9–17 years.

## Sampling technique and sample size

The sampling technique and sample size strategies are adapted from Chan *et al.*, [23,24] and Nguema *et al.*, [25]. Briefly, a two-stage random sampling was used at district level for selection of schools and children to participate in the study in order to produce precise estimates of prevalence [5,6]. At first, each district was subdivided into one to three different ecological zones based on proximity to inland temporary and permanent water bodies (near, less than 1km, medium-1-5km and far, greater than 5km). An ecological zone was defined as areas located within a similar distance from bodies of water within a locality [24]. Then, from each ecological zone, schools were sampled and secondly, school children were sampled from the

 

selected schools. Considering the focal nature of schistosomiasis distribution, with communities living close to water bodies having the highest risk of infection, only schools located close to water bodies were selected for the study [26]. The WHO recommends that for surveys aimed at assessing the need for control measures, a sample size of 200–250 individuals is adequate sample for each ecologically homogenous areas in order to evaluate prevalence and intensity of infection [27,28]. From each school, randomly selected 250 pre-school and school aged children were selected from the list of registered children in each class attendance book (at least 50 children from 5 classes) on the day of sample collection. A total of 22,000 pre-school and school aged children were selected for the study. The sampling technique are described elsewhere [25,29].

## Data collections

**Questionnaire.** A pre-tested questionnaire was used to collected pre-school and school aged children demographic information, mainly age, sex, class, school name, district and region and participation in mass drug administration. The questionnaire for children aged 3–6 years was administered to their caregivers (parents/guardians).

**Parasitological examination of *Schistosoma haematobium*.** On the day of urine sample collection, a single urine sample was collected from each participating child (collected after exercising and between 10:am-2:00pm). Collected urine samples were examined visually for presence of macrohaematuria using a colour chart and using reagent strips (Hemastix, Siemens healthcare Diagnostics GmbH, Germany) for micro-haematuria. The results of micro-haematuria were recorded as positive and negative.

Quantification of *S. haematobium* eggs followed the method described by WHO [13]. Collected urine samples were rigorously shaken and 10mls of urine were drawn using a 10mls plastic syringe from each urine sample and pressed through a polycarbonate filter [30]. All urine filters were microscopically examined after being stained with a drop of Iodine for presence of *S. haematobium* eggs by two medical laboratory technicians. The two technicians examined the slides independently and counted all visible *S. haematobium* eggs and recorded the results in the field log book. For quality assurance, during the fieldwork, 20% of the positive and negative microscopic slides were re-examined by third laboratory technicians.

**Geographical distribution of *Schistosoma haematobium* infection.** To determine the geo-prevalence distribution of *S. haematobium*, positions of all primary schools involved in the study were mapped using a Garmin hand-held Geographical Positioning System Unit (Trimble Navigation Ltd, California) [31]. All the GPS data were downloaded using the Arc-View software and analysis and the prevalence maps were generated using the ArcView 9.2 software (Environmental System Research Institute, Inc, Redlands, CA). The shapefiles/basely files for the maps were obtained from https://gadm.org/download_country.htmlPerhaps. Specific for Tanzania, shape files were downloaded from https://gadm.org/maps/TZA_1.html. Generated maps categorized prevalence levels based on WHO [32] categories (0%, 0.1–10.0%, 11–50% and ≥50%) as shown in **Fig 2** (https://gadm.org/maps/TZA_1.html).

**Data analysis.** All the analysis was performed using Stata version 15 (StataCorp, 2017, Stata statistical software, College Station, TX: StataCorp) and all statistical analysis with $P<0.05$ were considered statistically significant. A total of 20,389 pre-school and school aged children met the inclusion criteria (having complete data). *S. haematobium* infection was defined as presence of eggs in the urine filters examined under light microscopy. Geometrical mean eggs counts were estimated and comparison of the geometrical mean eggs counts between males and female or age groups was done either using t-test or ANOVA, depending on the number of comparison groups. Infection intensity was classified into two categories as

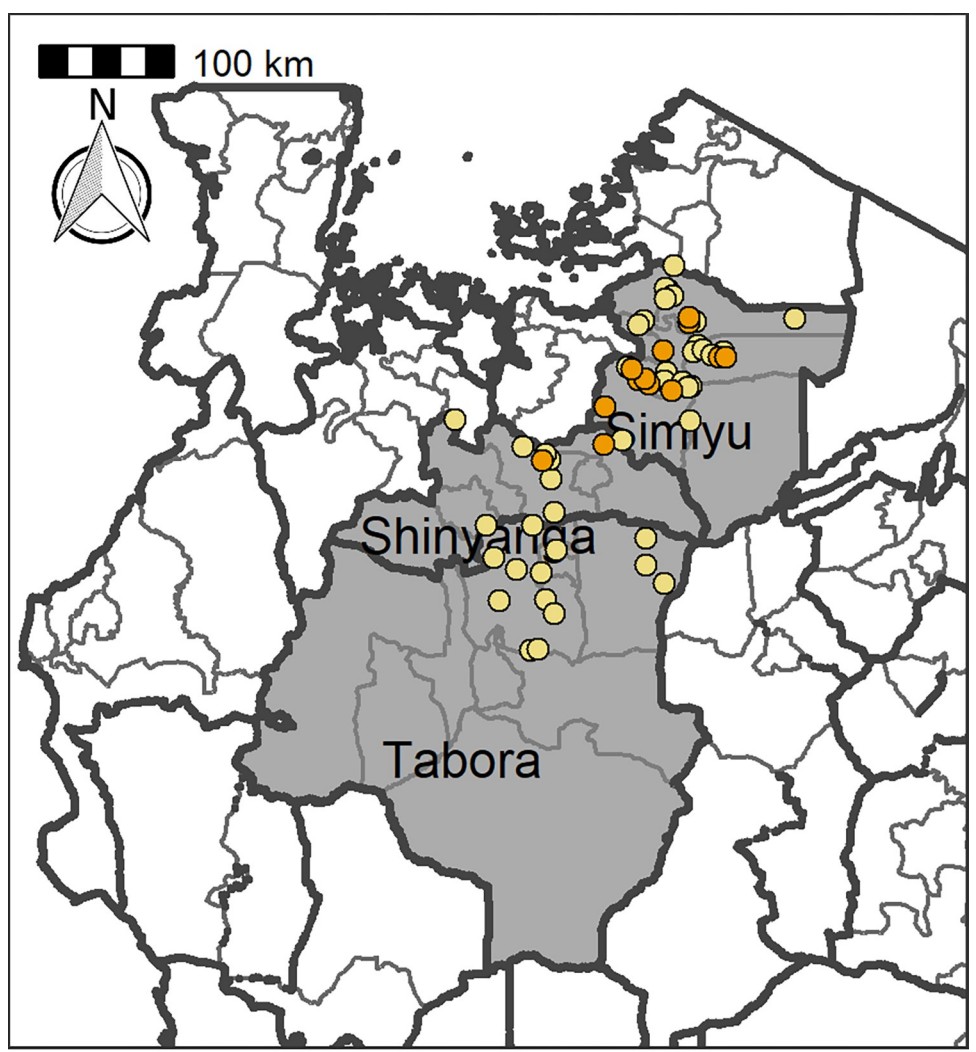

**Fig 2. Prevalence of *Schistosoma haematobium* among pre-and-school aged children in selected schools of Bariadi, Itilima, Shinyanga rural and Nzega districts, north-western Tanzania (credit: https://gadm.org/maps/TZA_1.html).**

per WHO recommendation [33] (i) light infection (<50 eggs/10ml of urine) and (ii) heavy infections ($\leq$ 50 eggs/10mls of urine). The association between *S. haematobium* infection (binary outcome) and explanatory categorical or continuous variables was assessed using bivariate and multivariate logistic regression. Similar approach was used to analyse for micro-haematuria and macrohaematuria data to identify the predictors of microhaematuria in the study group.

Analysis of the sensitivity and specificity of the reagent urine strip (indirect method) method was calculated as described elsewhere [11,34]. The sensitivity, specificity, negative and positive predictive values of the urine reagent strip were determine using egg microscopy, the urine filtration technique as the gold standard/the reference test. The sensitivity of the reagent

strip was calculated as proportion of positives that were correctly identified when compared to the reference test [11]. Sensitivity was calculated using the formula: -

**Sensitivity** = Number of true positives (TP)/(number of TP + number of false negatives (FN)).

Specificity of the reagent urine strip was calculated as the percentage of negative individuals correctly identified as negative compared to the reference test [11]. Specificity was calculated using the formula: **Specificity** = Number of true negatives (TN)/(number of TN+ number of false positives (FP)).

In addition, calculated were the Positive Predictive Value (PPV), i.e. the proportion of positive test results that are truly positive, and negative predictive value (NPV), i.e. the proportion of negative test results that are truly negative. These were calculated using the formulas: (i) PPV = TP/(TP + FP) and (ii) NPV = TN/(TN + FN) [35]. All the analysis were done using Stata version 15, using the Stata command diagt.

## Results

A total of 20,389 pre-school and school aged children from 88 primary schools located in four districts of north-western Tanzania participated in the study. Overall, 48.9% (9,974/20,389) and 51.1% (1,0415/20,389) were male and female respectively. The mean age of the study participants was 9.25 ± 2.44 years. Table 1 shows age, sex and the districts involved in the study.

### Prevalence and infection intensities of *Schistosoma haematobium*

A total of 1,514 pre-school and school aged children had eggs of *S. haematobium* in their urine samples diagnosed using urine filtration technique, giving an overall prevalence of 7.4%,95% CI:7.0–7.7 (1,514/20389). In relation to sex, male participants had the highest prevalence of *S. haematobium* than female participants (5.9% *versus* 9.01%, $\chi^2$ = 71.6161, *P*<0.001). However, there was no age differences in prevalence of *S. haematobium* ($\chi^2$ = 2.6759, *P* = 0.44). Fig 3 present prevalence of *S. haematobium* in different age groups in males and females.

The overall Geometrical Mean Eggs counts was 15.78eggs/10mls of urine, 95%CI: 14.8–16.8, (Range: 1–608 eggs). Male participants had higher geometrical mean eggs counts than female participants (13.8eggs[95%CI:12.6–15.2]/10mls of urine *versus* 17.3eggs(95%CI:15.9–18.7)/10mls of urine, t = -6.9364, *P*<0.001)1), but there were no age groups differences in mean eggs intensities (F = 1.24, *P* = 0.29). Based on intensity of infection categories, 82.3% and 17.7% of infected children had light and heavy infection intensities. Most heavy infections were observed in male participants (14.9% *versus* 19.5%). In relation to age groups, 18.5% of the children in the age group 11–15 years had heavy infection intensities.

**Table 1. Age, sex and distribution of age in each study district, north-western Tanzania.**

| Variable | Age in years (age groups) | | | |
|---|---|---|---|---|
| | 1–5 | 6–10 | 11–15 | 16–19 |
| Female | 682 (53.8%) | 6,779 (54.2%) | 2950 (45%) | 4 (8.3%) |
| Male | 586 (46.2%) | 5741 (45.8%) | 3603 (54.9%) | 44 (91.6%) |
| **Districts** | | | | |
| Bariadi | 270 (21.3%) | 2917 (23.3%) | 2119 (32.3%) | 20 (41.6%) |
| Itilima | 599 (47.2%) | 4935 (39.4%) | 2033 (31%) | 6 (12.5%) |
| Nzega | 317 (25%) | 3290 (26.3%) | 2082 (31.8%) | 22 (45.8%) |
| Shinyanga rural | 82 (6.5%) | 1378 (11%) | 319 (4.0%) | 0 (0.0%) |

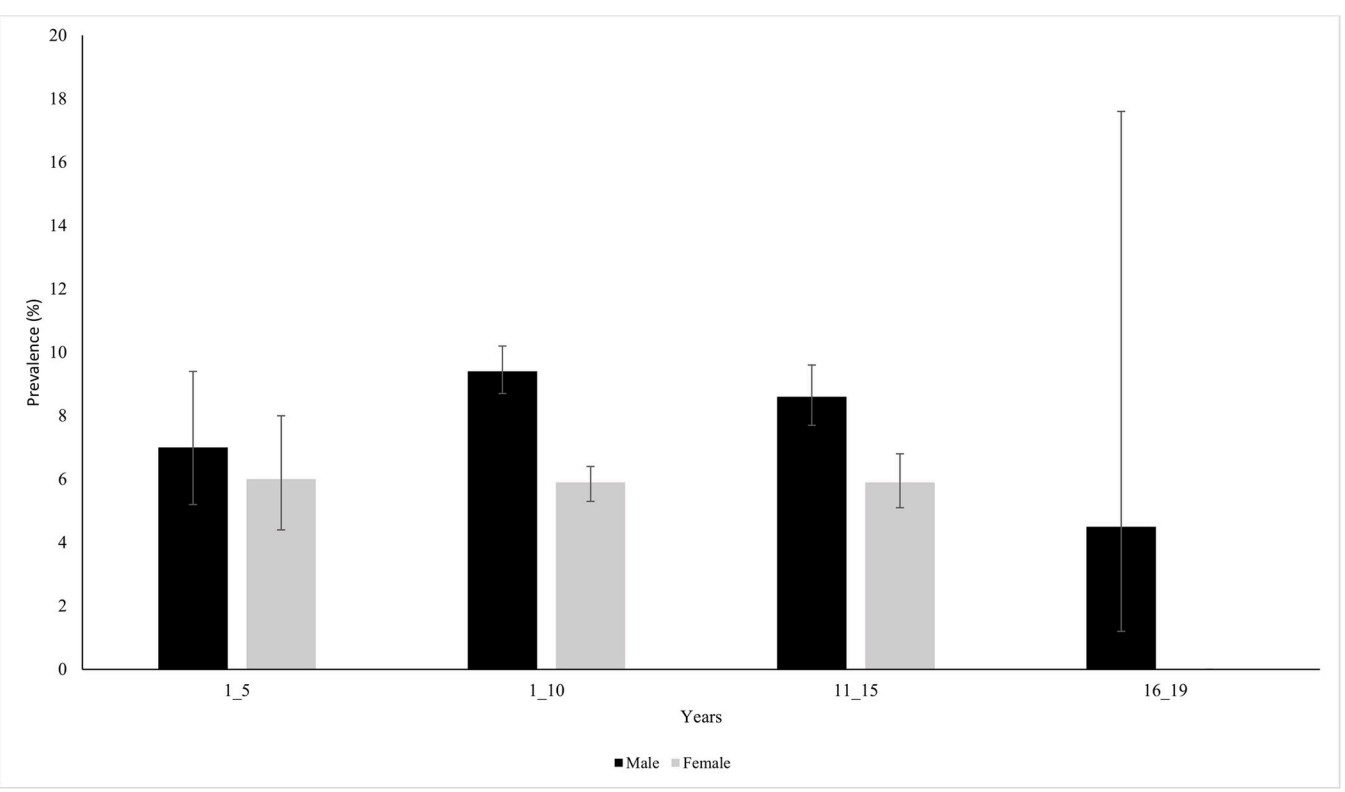

**Fig 3. Prevalence of *Schistosoma haematobium* categorised by age and sex in infected pre-school and school aged children in north-western Tanzania.**

## Prevalence of microhaematuria and its performance in detecting *S. haematobium* infection

The overall prevalence of macrohaematuria and microhaematuria were 0.3% (95%CI:0.3–3.5) and 9.3% (95%CI: 8.9–9.7) respectively. For microhaematuria, male participants had higher prevalence than female (7.7% *versus* 10.9%, $\chi^2$ = 64.1069, $P < 0.001$). However, there was no age group difference in prevalence of microhaematuria ($\chi^2$ = 0.9486, $P = 0.62$).

Among 1,514 urine samples which were identified to have *S. haematobium* eggs, 1,480 (97.8%) were microhaematuria positive and among 18,475 *S. haematobium* negative urine samples, 417 (2.2%) were microhaematuria positive. Compared with *S. haematobium* egg negative children, egg positive children were more likely to be diagnosed with microhaematuria (OR = 1926.8,95%CI: 1352.9–2743.9, $P < 0.001$). The odd of having microhaematuria positive results were mainly associated with having light and heavy infection intensities and living in the study districts (Bariadi, Itilima and Shinyanga rural districts) compared to living at Nzega district, the district with the lowest prevalence. **Fig 4** present the associations between microhaematuria, infection intensities and study districts.

**Table 2** shows the specificity and sensitivity of the urine reagent strip methods (urine dipstick). The overall sensitivity of the reagent urine strip method was 78% (95%CI: 76.1–79.9) and specificity was 99.8% (95%CI: 99.7–99.9).

## Predictors of *Schistosoma haematobium* infection

In bivariate analysis, being male, districts (Bariadi, Itilima and Shinyanga rural), macrohaematuria and microhaematuria predicted *S. haematobium* infection. In multivariate analysis, the

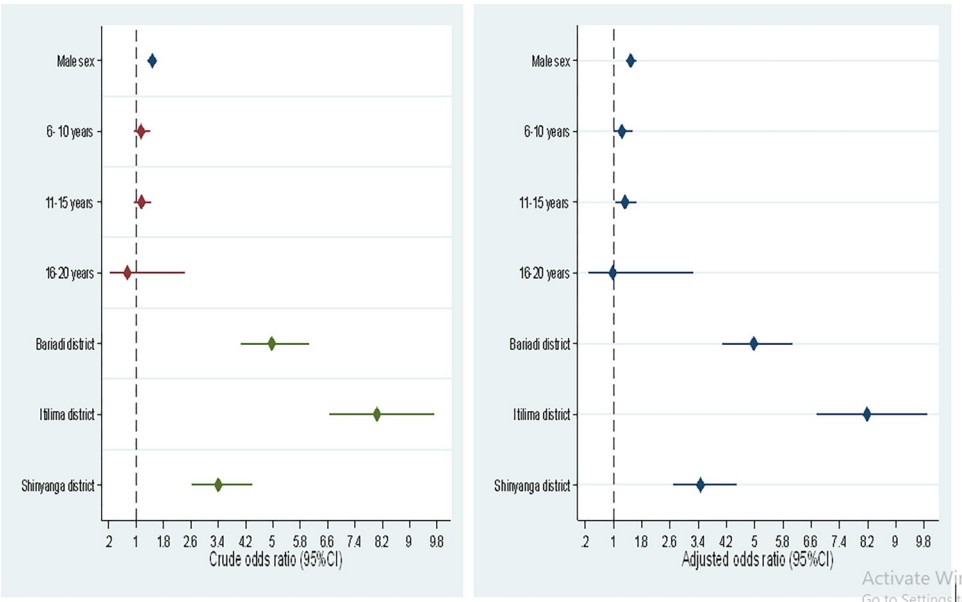

**Fig 4. Factors associated with microhaematuria among pre-school and school aged children in north-western Tanzania.**

main predictors of *S. haematobium* infection were being male (aOR = 1.4,95%CI:1.1–1.7, *P*<0.003), microhaematuria (aOR = 1623.8,95%CI:1138–2315.7, *P*<0.001), living in Bariadi (aOR = 3.2,95%CI:2.1–4.6, *P*<0.001), Itilima (aOR = 5.0,95%CI:2.1–4.6, *P*<0.001) and Shinyanga rural (aOR = 2.5,95%CI:1.4–4.1, *P*<0.001) districts remained associated with *S. haematobium* infection. Table 3 shows predictors of *S. haemaobium* infection.

## Prevalence of *Schistosoma haematobium* by districts

The study included four districts, namely Bariadi, Itilima, Nzega and Shinyanga rural. There was a significant variation in prevalence of *S. haematobium* between the four districts ($\chi^2$ = 646.7403, *P*<0.001), with Itilima district having the highest prevalence (12.7%) and Nzega district having the lowest prevalence (1.1%). Shinyanga rural and Bariadi districts had a prevalence of 5.1% and 7.6% respectively. The prevalence of *S. haematobium* varied significantly between schools located in the same ecological zone. At Itilima district, the prevalence at school levels ranged from 0.2–8.5%, at Bariadi (1.2–13.8%), Shinyanga rural (1.1–27.5%) and Nzega (0–15.8%). At Itilima districts, in all the 23 schools involved in the study, the prevalence of *S. haematobium* was <10%. At Bariadi district, 19 schools had prevalence of *S. haematobium* >10% and four schools had prevalence <10% (Masewa B (10.9%), Mwagabate (12.6%), Ibulyu (13.4%) and Nyamikoma (13.7%). At Nzega district, out of 24 schools involved in the study, only two [Nkindu-11.1% and Ifumba-15.8%] had prevalence of *S. haematobium* above

**Table 2. Sensitivity and Specificity of urine reagent strip methods for *S. haematobium* diagnosis among pre-school and school aged children in north-western Tanzania using urine filtration technique as a reference.**

| Diagnostic test | Sensitivity | Specificity | PPV | NPV |
|---|---|---|---|---|
| Urine reagent strip | 78% (95%CI:76.1–79.9) | 99.8% (95%CI:99.7–99.9) | 97.8% (95%CI:96.9–98.4) | 97.8% (95%CI: 97.6–98.0) |

**Key:** *PPV-Positive Predictive Values * NPV- Negative Predictive Values

**Table 3. Predictors of *Schistosoma haematobium* infection among pre-school and school aged children in four districts of north-western Tanzania.**

| Variable | Unadjusted OR | | | Adjusted OR | | |
|---|---|---|---|---|---|---|
| | OR | 95%CI | *P*-value | aOR | 95%CI | *P*-value |
| **Sex** | | | | | | |
| Female | 1 | | | 1 | | |
| Male | 1.5 | 1.4–1.8 | 0.001 | 1.4 | 1.1–1.7 | 0.003 |
| **Age groups in years** | | | | | | |
| 1–5 | 1 | | | 1 | | |
| 6–10 | 1.2 | 0.9–1.5 | 0.2 | 1.3 | 0.8–2.1 | 0.2 |
| 11–15 | 1.2 | 0.9–1.5 | 0.2 | 1.3 | 0.7–1.8 | 0.5 |
| 16–20 | 0.6 | 0.1–2.6 | 0.5 | 0.9 | 0.1–10.8 | 0.9 |
| **Macrohaematuria** | | | | | | |
| No | 1 | | | | | |
| Yes | 60.0 | 30.1–119.7 | 0.001 | ------ | ------- | --------- |
| **Microhaematuria** | | | | | | |
| No | 1 | | | 1 | | |
| Yes | 1926.7 | 1352.9–2743.9 | 0.001 | 1623.8 | 1138.6–2315.7 | 0.001 |
| **District** | | | | | | |
| Nzega | 1 | | | 1 | | |
| Bariadi | 7.3 | 5.6–9.6 | 0.001 | 3.2 | 2.1–4.6 | 0.001 |
| Itilima | 12.9 | 10.0–16.7 | 0.001 | 5.0 | 3.5–7.3 | 0.001 |
| Shinyanga | 4.8 | 3.5–6.7 | 0.001 | 2.5 | 1.4–4.1 | 0.001 |

10%. Four schools [Mwasingu-20.9%, Mahembe-21.9%, Ng'homango-21.9% and Ikinwama-noti-27.5%] out of eight schools at Shinyanga rural district had prevalence of *S. haematobium* above 10%. In relation to intensity of infection, at Bariadi district, 75.4% and 24.6% of the children who egg positive results had light and heavy intensity of infection respectively. At Itilima district, 84.6% and 15.5% had light and heavy intensity of infection whereas at Nzega district, 85.7% and 14.3% had light and heavy intensity of infection. Lastly, 86.8% and 13.2% of the children had light and heavy intensity of infection at Shinyanga rural district.

Comparing the current findings and that of the earlier mapping survey in 2005/06 [10], clearly, there was significant decline in prevalence of *S. haematobium* in each of the study districts. During the 2005/06 mapping exercise, Itilima district was part of the Bariadi district, thus no comparison data for this district. **Fig 5** shows the prevalence of *S. haematobium* in each district prior to MDA in 2005/06 and 15 years after repeated rounds of MDA.

## Discussion

Re-assessment of *S. haematobium* infection levels after 15 years of MDA are an essential prerequisite for planning and implementation of cost-effective follow-up MDA rounds. Here, we present the results of the re-assessment of *S. haematobium* infection in four districts of north-western Tanzania. In general, this cross-sectional survey demonstrates that after a 15 years of mass drug administration in Tanzania, *S. haematobium* continues to be a public health concern among pre-school and school aged children in the study districts. The results confirm further a geographical variation in prevalence of *S. haematobium* between districts and schools located in the same district, as observed in the past 15 years [10]. Furthermore, the results confirm a decline in the prevalence of *S. haematobium* infection in the four districts, from an average prevalence of 20.7% in 2005/2006[10] prior to the start of national wide mass drug administration to 7.4% in 2019/20. The decline in prevalence is also reflected in individual

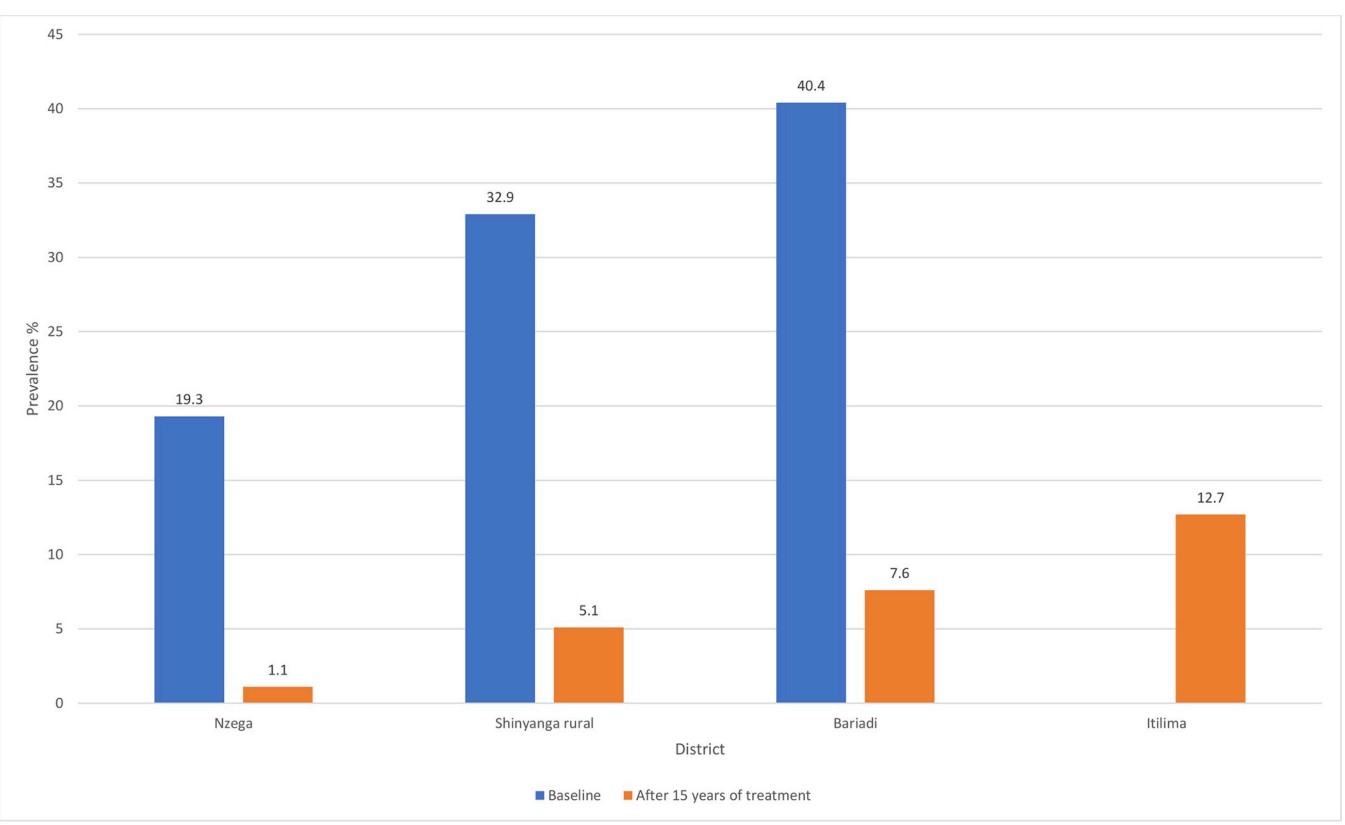

**Fig 5. Changes in prevalence of *Schistosoma haematobium* after 15 years of repeated rounds of mass drug administration in four districts of north-western Tanzania.**

districts, with Nzega district observed a significant decline from 19.3% in 2005/06[10] to 1.1% in 2019/20. Despite the observed decline, our findings demonstrate an existence of pockets for *S. haematobium* transmission in some district. Despite several rounds of MDA over a decade, the urine reagent strip(s) remained as a useful and basic diagnostic method for rapid assessment of *S. haematobium* in endemic areas. Its performance was influenced by intensity of *S. haematobium* infection and living in the study districts.

The current findings confirm the results of the previous mapping in the past 15 years which indicated that the four districts had low to moderate endemicity for *S. haematobium* infection [10], with the prevalence and intensity of infection varies from one district to another. In north-western region, transmission of *S. haematobium* is known to occur in Sukuma land south and southeast of the Lake Victoria, outside the lake basin [21,36]. In the past 15 years, the prevalence of *S. haematobium* infection have significantly declined in all the four districts involved in the study compared to the baseline mapping results of 2005/06 [10]. Partly, the observed decline can be explained by the intensive repeated rounds of mass preventive chemotherapy implemented in these districts in the past 15 years. It is worthwhile to note that not all districts included the current study achieved a similar decline in prevalence of *S. haematobium* infection. Of all the four districts, Nzega districts achieved the highest decline, from 19.3% in 2005/06 to 1.1% in 2019/20. Some districts have remained with pockets of infections, especially at Bariadi and Itilima districts. Some schools in these districts recorded a prevalence of *S. haematobium* >10%, even though these schools have been part of annual MDA in the past 15 years. The geographical landscape of Itilima and Bariadi districts is characterised by black soil

with multiple in land natural water bodies which are the main source of water for domestic, recreational, agriculture and animals use. These water bodies maintain the transmission of *S. haematobium* throughout the year among community members [21,37]. This is different from Nzega and Shinyanga rural districts which have few inland water bodies, which dries out during the dry season and affects the transmission of *S. haematobium* [21]. Partly, this could explain the variation in prevalence of *S. haematobium* observed between the districts.

These findings have implications on the planning and implementation of next rounds of MDA. The WHO have set criteria for schistosomiasis prevalence and morbidity control [38], with an ambitious goal of eliminating schistosomiasis as a public health problem (defined as <1% proportion of heavy intensity of infections) in all endemic countries by 2030 [39]. Based on this criterion, morbidity control can be achieved if the prevalence of heavy infection intensity with any schistosomes species is reduced to <5% [38] and for MDA, wherever the prevalence of schistosomiasis prevalence exceeds 10% [38]. Following this recommendation, majority of the pre-school and school aged children especially those living at Nzega district will not require every year MDA, the highest schools' prevalence was 15.1%% with almost 80% of the schools had prevalence <10%. Conversely, in other districts involved in the study, schools recording a prevalence of <10% may not need every year annual MDA. However, schools classified as moderate-risk (prevalence ≥10%—<50%) [32], which were few and located at Itilima, Bariadi and Shinyanga rural districts would require treatment once every two [2] years [38]. To significantly reduce prevalence and interrupt transmission in areas classified as moderate-risk areas, it will be important in the future to intensify intervention measures. Alternatively, there is need to design a well-focused treatment strategy focusing only in schools classified as moderate-risk schools. This proposed strategy should focus on designing a treatment strategy that will incorporate in transmission seasons of *S. haematobium* in local environment. Webbe suggested that in areas where a season with high transmission of *S. haematobium* is followed by a season with low transmission, PZQ treatment given at a defined time period will result in improved cures rates and interrupt transmission for long-period [40]. In north-western Tanzania, studies on the dynamic of transmission of *S. haematobium* have indicated that transmission is intense between February-March in which new cases of infections are detected in April/May [21] and between July-September in which new cases of infection are detected in September/October [36]. Authors suggested that for the MDA intervention to have long-term impact, preventive chemotherapy would best be administered in September and October [3,21]. At this time, apart from reducing morbidity, the intervention will lower egg output and hence reduce infection to intermediate host snails during the main transmission period [21,36]. This will minimise re-infection to treated children. To integrate control measures, other authors have suggested that molluscicides can be applied in focal areas of intense transmission between December/January to reduce snail population before the peak of transmission period occurs [21]. However, these recommendations have never been evaluated and needs to be tested in a well-designed study which will capture the transmission cycles in different geographical settings, especially at this time were repeated multiple rounds of mass treatment have significantly changed the epidemiology of the disease [7–9] and climate changes have changed the ecology of the disease [41].

## Performance of urine reagent strips and associated factors

Urine reagent strips and urine filtration techniques are the basic diagnostic techniques for diagnosis of *S. haematobium* infection [30]. The sensitivity of these techniques is affected or reduced in areas with low prevalence or following repeated mass drug administration due to declining in prevalence and infection intensity [12,42]. After a decade of MDA in north-

western Tanzania, it is important to assess its performance and factors which predict its positivity [11]. Overall, after a decade of repeated mass preventive chemotherapy, the performance of urine reagent strip is acceptable and the tool can continue to be used for diagnosis and monitoring the impact of treatment. The findings on the sensitivity and specificity of urine reagent strip were consistent with results of the previous studies [11,14,16,17]. A recent meta-analysis, reported sensitivities of urine reagent strips of 65% for light intensity of infection and 72% for post treatment groups [12]. The overall sensitivity of detecting *S. haematobium* cases in our present study was 78% and the specificity was 97.8%. This high sensitivity and specificity of the urine reagent strips were in line with similar study conducted in low and moderate transmission zones in Ethiopia [14], but higher than 71.6% and 69.7% reported among children and adult in Zanzibar [11]. In line with the findings of other authors [11], microhaematuria was strongly associated with *S. haematobium* infection, the likelihood of urine samples being microhaematuria positive increased significantly from light to heavy infection egg outputs. In addition, living in the study district, which are known to be endemic for urogenital schistosomiasis [2] increased the likelihood of having microhaematuria positive results. In general, urine reagent strips and urine filtration technique remain as valid diagnostic means for *S. haematobium* infection in epidemiology mapping surveys [12]. In addition, these tools are suitable means for monitoring and evaluation progress of preventive chemotherapy in areas with low and high transmission intensities [15,16], but for measuring changes in infection intensities, urine filtration technique which allow egg counting is suitable [30].

## Limitations

It should be noted that this present study was not conducted without any limitation, the parasitological results used to define the prevalence and intensities of infection are based on single sample collected from the study participant. This may have underestimated the true prevalence and intensities of infection. The cross-sectional nature of the study design could not provide data on causality. However, using this design, we are able to describe the prevalence and intensities of infection of *S. haematobium* after 15 years of MDA. Conversely, the lack of a third and highly sensitive diagnostic technique such as Polymerase Chain Reaction to validate the results of the urine reagent strips, urine filtration technique and the lack of baseline data on the performance of the urine reagent strip which would allow comparisons of its performance (before and after 15 years of MDA) limits the interpretation of the findings. Lastly, the data were collected between October and November, the period deemed to be a low transmission period [3,21]. Incidence mapping study to schools which had no cases and low prevalence (<10%) is going on. Nevertheless, the findings from the current study allows for the following conclusions and consideration.

## Conclusions and recommendations

The present study provides data on the geographical distribution of *S. haematobium* infection in four districts after a decade of mass drug administration. In general, the prevalence of *S. haematobium* infection have significantly declined in the four districts and partly, this can be attributed to repeated rounds of mass preventive chemotherapy implemented over the past 15 years. However, some schools still maintain the prevalence of infection >10%. These schools require an uninterrupted annual MDA or a focused treatment strategy which encompasses transmission season of *S. haematobium* in a local environment. It should be noted however, complementary measures which include health education, improving safe water supply and adequate sanitary facilities are recommend. Lastly, the urine reagent strips remain as a useful adjunct diagnostic test for rapid monitoring of urogenital schistosomiasis even after 15 years of repeated rounds of mass preventive chemotherapy.

## Acknowledgments

The authors would like to thank the study participants for their time and responses without which this study would not possible. We also thank the regional, district, schools, and village authorities where the study was conducted for granting the permission and providing every support to successfully conduct this study. **The views expressed in this publication are those of the author(s) and not necessarily those of the NIHR or the UK Department of Health and Social Care. Its contents are solely the responsibility of the authors and do not necessarily represent the official views of the supporting offices.**

## Author Contributions

**Conceptualization:** Humphrey D. Mazigo, Upendo J. Mwingira, Francesca Mutapi.

**Data curation:** Humphrey D. Mazigo, Maria M. Zinga, Cecilia Uisso, Paul E. Kazyoba.

**Formal analysis:** Humphrey D. Mazigo, Francesca Mutapi.

**Funding acquisition:** Humphrey D. Mazigo, Upendo J. Mwingira, Francesca Mutapi.

**Investigation:** Humphrey D. Mazigo, Maria M. Zinga, Cecilia Uisso, Paul E. Kazyoba.

**Methodology:** Humphrey D. Mazigo, Upendo J. Mwingira.

**Project administration:** Humphrey D. Mazigo, Cecilia Uisso, Paul E. Kazyoba.

**Supervision:** Safari M. Kinung'hi, Francesca Mutapi.

**Writing – original draft:** Humphrey D. Mazigo.

**Writing – review & editing:** Humphrey D. Mazigo, Upendo J. Mwingira, Maria M. Zinga, Cecilia Uisso, Paul E. Kazyoba, Safari M. Kinung'hi, Francesca Mutapi.

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
