## [Decision Letter · Decision Letter 0]

17 Nov 2021

Dear Prof. Mazigo,

Thank you very much for submitting your manuscript "Urogenital schistosomiasis among pre-school and school aged children in four districts of north-western Tanzania after 15 years of mass drug administration: geographical prevalence, risk factors and performance of haematuria reagent strips" for consideration at PLOS Neglected Tropical Diseases. As with all papers reviewed by the journal, your manuscript was reviewed by members of the editorial board and by several independent reviewers. In light of the reviews (below this email), we would like to invite the resubmission of a significantly-revised version that takes into account the reviewers' comments. 

This is an important study but there must be a significant effort on the part of the authors to address the various comment of the three reviewers. Two of the reviewers mentioned major reviews, please read their comments carefully, the work need to published, but the out put could be of considerable impact.

We cannot make any decision about publication until we have seen the revised manuscript and your response to the reviewers' comments. Your revised manuscript is also likely to be sent to reviewers for further evaluation.

Sincerely,

Clive Shiff

Associate Editor

Poppy Lamberton

Deputy Editor

This is an important study but there must be a significant effort on the part of the authors to address the various comment of the three reviewers. Two of the reviewers mentioned major reviews, please read their comments carefully, the work need to published, but the out put could be of considerable impact.

Reviewer's Responses to Questions

**Key Review Criteria Required for Acceptance?**

**Methods**

-Are the objectives of the study clearly articulated with a clear testable hypothesis stated?

-Is the study design appropriate to address the stated objectives?

-Is the population clearly described and appropriate for the hypothesis being tested?

-Is the sample size sufficient to ensure adequate power to address the hypothesis being tested?

-Were correct statistical analysis used to support conclusions?

-Are there concerns about ethical or regulatory requirements being met?

Reviewer #1: The manuscript is primarily descriptive, with no particular hypothesis put forth or tested. The authors do address whether or not reagent strips are useful for adjunct diagnosis for S. haematobium infection, which would be informative for other programs, but did not lay out a clear comparison of how "usefulness" was tested or determined.

Reviewer #2: Schistosomiasis still poses a huge public health concern in many sub Saharan countries. Many of these countries including Tanzania have implemented mass drug administration (MDA) targeting primary school children for the control. After a decade of national MDA in Tanzania the strategy and infection diagnostic tools have been evaluated to inform the next stage of the schistosomiasis control activities.The findings from the study provide an updated geographical prevalence giving insight on the planning of MDA. The prevalence of S. haematobium infection have significantly declined informing about the success of the programme. Appropriate design has been used to evaluate the success of the control programme. The results pointed to the success of the repeated rounds of mass drug administration. The urine reagent strips remain as a useful diagnostic tool for rapid monitoring of urogenital schistosomiasis in areas with moderate to high prevalence. The study set out clear objectives that have been addressed and the also the study got thorough ethical approval before implementation.This undertaking permitted to gauge the success of the MDA programme and also assisted mapping the distribution after several rounds of MDA. The way the data have been presented provides an exceptional example to be applied in different settings and map the way forward on the control exercise.

Reviewer #3: Yes

**Results**

-Does the analysis presented match the analysis plan?

-Are the results clearly and completely presented?

-Are the figures (Tables, Images) of sufficient quality for clarity?

Reviewer #1: --what was the relationship between the 86 schools surveyed in 2005/2006 and 88 schools surveyed in 2019? A great deal of overlap or were they mostly different schools. Rather than one map combining all the schools from the baseline (Figure 1) and then district level maps (Figs 3-6) for the follow up survey, it would be more useful to have side by side baseline and 2017 maps for each district so the changes can be more readily seen. As part of these paired maps, the number of rounds and intervals of MDA should be provided for each district. Figure 2 is not very useful at all and should be deleted. 

--the authors stated that the surveyed schools were classified according to ecologic zone (distance from water body) but no analysis based on ecologic zone is presented.

--similarly the authors state that a questionnaire was used to collect demographic information, water source, knowledge of open water sources, symptoms, and participation in MDA; however, none of these data were analyzed or presented in the results. Only differences in prevalence according to districts were presented but there was no indication of why some districts were higher or lower than others.

--results of reagent strips for microhaematuria were recorded semi-quantitatively but not reported or compared to eggs counts. One useful thing this paper could present is the relationship between infection intensity based on egg counts and semi-quantitative microhaematuria--is it the same at baseline and post-several rounds of MDA or does having treatment rounds change the dynamic? This information would make the paper more attractive to a wider audience. 

--the way Table 2 is laid out and labeled suggests that it is urine filtration that is being evaluated rather than the urine reagent strip.

--Figure 10 should be a bar graph rather than a line graph as there is no continuum between the districts like there would be if time were being plotted. Where are the 2005/2006 data from Itilima? Line 309 suggests 15 years of MDA but the figure legend states it is a decade. How many rounds of MDA were delivered in each district?

--the authors state that some districts have hotspot areas but do not define what constitutes a hotspot or what might be different between hotspot and non-hotspot areas.

Reviewer #2: The results are outlined appropriately and following the objectives set out to answer. The figures are clear and well presented with the exception of Figure q10 that would be more appropriate presented as a bar graph instead of lines. The lines do not represent the linkage between the different points. The data availability provides the opportunity to create big data set and reveal the trends combining with other countries in sub Sahara.

Reviewer #3: Yes

**Conclusions**

-Are the conclusions supported by the data presented?

-Are the limitations of analysis clearly described?

-Do the authors discuss how these data can be helpful to advance our understanding of the topic under study?

-Is public health relevance addressed?

Reviewer #1: As a purely descriptive study that there are more infections in some district than others and that prevalence differs by age and sex, the conclusions are supported by the data. However, the criteria that many rounds of MDA does not change reagent strip performance is not directly evaluated other than showing the sensitivity and specificity data. But whether sensitivity was higher pre-MDA is not addressed.

Reviewer #2: The conclusion are well supported by the data and the presentation is clear and showing the trends. the authors mentioned the study limitations while the analysis makes it clear on the current trends in which mass treatment is the mainstay of control initiatives in the sub Sahara region. The comparison to the original surveys before the MDA initiation is important and showing the progress made so far afte several rounds of MDA. The data and analysis gives an opportunity to evaluate the progress and also see the gaops in the control initiatives.

Reviewer #3: Yes

**Editorial and Data Presentation Modifications?**

Reviewer #1: The English is good enough that the intentions of the paper are easily understood but it has numerous subject/verb agreement errors, missing articles, and simply oddly worded sentences that are not clear (e.g. lines 56-57, 102-106, 119, 121, that it would greatly benefit from copy editing.

line 93--urine filtration for eggs is the standard method for S. haematobium diagnosis but it is not a "gold" standard.

line 202--mentions a third laboratory technician but it was never disclosed that 2 read each slide to begin with. Is that correct?

line 257 says the majority of male patients had heavy infections but this is not true as only 19.5% did. However, the majority of heavy infections were in males is a true statement. There a simple errors like this throughout the paper that should be corrected.

Reviewer #2: The data is professionally handled and given in a clear manner. However I noticed the mode of presentation of figure10 may require revision to show as a bar graph instead of lines joining different study sites. The same data can be appropriateiy presented as bars.

Reviewer #3: • Line 327: Is the 21.7% mentioned here correct? Should it be 20.7%?

• There are a few minor language corrections, e.g.: line 27: “Cross-sectional study” should be “A cross-sectional study”

• Line 38: “Low” should be “Light”

• Line 88: “majority” should be “the majority”

• Line 101: “needs to re-assessment” should be “needs re-assessment”

• Line 102: “ad” should be “and”

• Line 141: “inland” is repeated

• Line 256: “Majority” should be “More”.

• Line 270: “were likely” should be “were more likely”

**Summary and General Comments**

Reviewer #1: As presented, the study is primarily descriptive and not very generalizable to other settings. The frequency of MDA distribution is not clear (15 years in the title but a decade in line 25) and without a clear history for each district it is not possible to determine whether hotspots are the result of some ecological difference or simply reflect more, or more consistent, MDA is some areas compared to others. However, these same data could be used to develop a much more compelling manuscript. For example, as mentioned in the results section, do multiple rounds of MDA in a district change the relationship between egg count and level of microhaematuria?

Reviewer #2: there general minor corrections need to be observed at he following page points:

page 27A cross sectional study

page 116 form to take to their parents

page 119 are kept in closed cabinet

page 121 were treated using pZQ

page 141 remove man-made inland and natural

page 198 please check the pore size if not in micro metres

page 277 line should be linked to line 278

page 389 its performance

page 434 this study would not be ....

Change figure10 lines to stand alone bars

Reviewer #3: This is a really rich sample of over 20,000 children from four districts, with 250 children from each school. This area has been under treatment for a long time, and so it is interesting to see the level of infection in this area, and any change from baseline. 

Given the size of the sample size, I think there are a number of additional useful analysis that could be carried out. In particular looking at the distribution of infection by sub-district, and looking at school-level hotspots of infection.

Comments

• Given the sample size, and the WHO push for sub-district level implementation, is it possible to define results (and maps) by sub-district? That would help to fine-tune the implementation approach

• I didn’t see any analysis of the difference in infection markers between the three ecological zones (distance from lake). Can this be added?

• The authors talk about many hot-spots at the school-level remaining. How many of these fall into that category?

• I would say there is only a modest reduction in prevalence from baseline, which in a sense is disappointing. However, macrohaematuria is very low which is reassuring. As a more direct measure of morbidity it is more reassuring to see this fall than prevalence. 

• How many of the districts / sub-districts have reached the metric of <1% heavy infections which indicates elimination as a public health problem?

• Given that there have been so many rounds of treatment in this area, are there conclusions that can be drawn about the usefulness of the MDA approach? E.g. that it can be used to suppress infection but cannot take us to elimination. 

• Check author affiliations. Is Cecilia Uisso affiliated to RTI?

• It is a long time since baseline. Were any intensity measures available from baseline? Have there been any other surveys between the two time-points?

• Aims: “Thus, the overall aims of the present study were to (i) determine the prevalence ad intensity of infection of S. haematobium, prior to implementation of mass preventive chemotherapy to allow monitoring of the impact of treatment on infection prevalence and intensities”. Consider re-wording, it’s not prior to implementation.

• Line 124: you say that these areas are highly endemic for SCH, but pre-MDA mapping categorized them as low to moderate endemicity. Which is correct?

• Lines 208-209: “Generated maps categorized prevalence levels based on WHO categories (0%, 0.1-10.0%, 10.1-20.0%,>20%)” Are they the cut-offs? I think for SCH, they are 0%, 10%, and 50%

• Many fewer children were sampled in Shinyanga district – why is that?

• Lines 252 – 254. If overall intensity was 15.78 e/10ml, I don’t understand how males can be 3.26 e/10ml and females 1.68 e/10ml. I think for female instead of “1.68 +/- 13.29” it should be “13.29 +/- 1.68”

• Figure 9 – in multivariate analysis, was there no longer an association between haematuria and being infected with S.haematobium? That would be an interesting finding.

• Figure 10: Recommend redrawing as bar chart, rather than line chart.

• Titles for Figures 3-6 need more information. Are these the results from the current study?

PLOS authors have the option to publish the peer review history of their article (what does this mean?). If published, this will include your full peer review and any attached files.

Reviewer #1: No

Reviewer #2: Yes: Professor Takafira Mduluza

Reviewer #3: Yes: Michael French
---

## [Editor Report · Decision Letter 1]

20 Sep 2022

Dear Prof. Mazigo,

We are pleased to inform you that your manuscript 'Urogenital schistosomiasis among pre-school and school aged children in four districts of north western Tanzania after 15 years of mass drug administration: geographical prevalence, risk factors and performance of haematuria reagent strips' has been provisionally accepted for publication in PLOS Neglected Tropical Diseases.

Best regards,

Clive Shiff

Academic Editor

Poppy Lamberton

Section Editor

I have reviewed the comments made by reviewers and the responses made by the authors, and an in agreement of the responses and changes. The paper can progress to publication

---

## [Editor Report · Acceptance letter]

7 Oct 2022

Dear Prof. Mazigo,

We are delighted to inform you that your manuscript, "Urogenital schistosomiasis among pre-school and school aged children in four districts of north western Tanzania after 15 years of mass drug administration: geographical prevalence, risk factors and performance of haematuria reagent strips," has been formally accepted for publication in PLOS Neglected Tropical Diseases.

Best regards,

Shaden Kamhawi

co-Editor-in-Chief

Paul Brindley

co-Editor-in-Chief
